# Dietary Canthaxanthin Supplementation Promotes the Laying Rate and Follicular Development of Huaixiang Hens

**DOI:** 10.3390/biology12111375

**Published:** 2023-10-27

**Authors:** Zhuangzhi Zhao, Jiang Wu, Yuan Liu, Yijie Zhuang, Haoguo Yan, Mei Xiao, Li Zhang, Lilong An

**Affiliations:** Department of Animal Science, College of Coastal Agricultural Sciences, Guangdong Ocean University, Zhanjiang 524088, China; 15291237200@163.com (Z.Z.); wuj@gdou.edu.cn (J.W.);

**Keywords:** canthaxanthin, Huaixiang chicken, laying rate, follicular development, reproductive hormones

## Abstract

**Simple Summary:**

Huaixiang chicken is a famous local broiler in China. It has excellent meat quality, but its disadvantage is a low laying rate. At present, improving the laying rate of poultry via nutritional regulation has attracted the extensive attention of researchers. Canthaxanthin(CX) is a ketocarotenoid, which widely exists in marine animals, algae, and a few terrestrial plants. It has been used in poultry production. In this study, dietary CX supplementation is found to increase laying rates, promote ovulation and maintain reproductive hormone levels by increasing serum and ovarian antioxidant levels. It is suggested that CX can be used to improve the laying rate of Huaixiang broiler breeders.

**Abstract:**

Canthaxanthin(CX) is a ketocarotenoid, which is widely used in poultry production as a lipophilic antioxidant. Huaixiang chickens are a local breed in China famous for their excellent meat quality; improving their laying rate via nutritional regulation has attracted extensive attention. The aim of this study was to evaluate the effects of dietary CX on the laying rate and follicular development in Huaixiang hens. A total of 180 Huaixiang hens were randomly divided into five groups with six replicates, and six chickens per replication. The control group (CON) were fed a basal diet, and the treatment group (NT) were fed a basal diet supplemented with 4, 6, 8 and 10 mg/kg CX. All chickens were 26 weeks old, living at an average environmental temperature of 25 ± 2 °C with a relative humidity of 65–75%. The results showed that supplementing the CX improved the laying rate and large white follicles (LWF) number (*p* < 0.05) and increased the concentration of reproductive hormones (LH, FSH, E_2_ and Prog) (*p* < 0.05), and the basal diet supplemented with 6 mg/kg CX worked best. Moreover, CX could increase the activities of antioxidant enzymes SOD and GSH-Px (*p* < 0.05) and reduce the content of the lipid peroxidation product MDA in Huaixiang chickens (*p* < 0.05); again, 6 mg/kg CX was best. In conclusion, dietary CX had positive effects on the laying rate, ovarian structure, reproductive hormone secretion, follicle development, and the antioxidant capacity of Huaixiang hens, and 6 mg/kg CX was recommended to be added to the diet of Huaixiang chickens.

## 1. Introduction

Breeding hens producing sufficient numbers of high-quality eggs and maintaining a good physiological condition are essential for the breeding of good poultry breeds [1,2]. The laying rate is a central issue in the production of commercial layers and breeding hens, and the effective induction of follicular development in laying production could be a promising strategy to improve the laying rate [1,3]. The hypothalamus of the poultry receives stimulation to secrete gonadotropin-releasing hormone (GnRH), which regulates the secretion of prolactin (PRL), follicle-stimulating hormone (FSH), and luteinizing hormone (LH) by the pituitary through body-fluid regulation to regulate poultry egg production. In addition, the biological effects of FSH and LH secreted by the pituitary gland are triggered by corresponding specific receptors in the ovary to regulate ovarian and follicle development [4,5,6,7].

Serum reproductive hormone level is an important index for evaluating egg-laying poultry [8]. The hormones secreted and released by the poultry HPG axis include E2, P4, FSH and LH, among which FSH and LH can promote the development and maturation of follicles and the POF granular layer, while FSH and LH can promote the secretion and release of progesterone and estradiol by follicle sheath cells and granular cells [6,9]. Studies have shown that the interaction between reproductive hormones and their receptors plays an important role in poultry reproductive activities. FSHR and LHR promote granulosa cell differentiation and dominant follicle selection in poultry, regulate the growth and development of ovaries and follicles, produce steroid hormones, and promote the synthesis of progesterone. ER and PR are of great significance in promoting ovarian development, maintaining and initiating egg production in female poultry [10]. The demand for egg production in poultry will also accelerate the body’s metabolism and increase oxygen demand, inducing an increase of reactive oxygen species (ROS) in ovarian tissue [11,12]. In severe cases, it will break the oxidation–antioxidant balance of the body, resulting in oxidative stress in the poultry’s body, inducing the oxidative damage of organs, and reducing the antioxidant performance of the body. It is manifested as an increased content of malondialdehyde (MDA), decreased activity of antioxidant enzymes superoxide dismutase (SOD) and glutathione peroxidase (GSH), and a decreased total antioxidant capacity (T-AOC) [11,13,14].

Nutritional regulatory substances in the diet can affect egg quality and later embryonic development; especially, the efficacy of natural antioxidants has attracted wide attention [3,15]. Among them, carotenoids, mainly existing in the chloroplasts and chromatin of photosynthetic organisms such as plants and algae, are widely used in animal diets to promote growth and reproduction as a result of their good antioxidant and free radical-scavenging abilities [16,17]. Canthaxanthin (CX) is a lipophile antioxidant that acts as a dike to carotenoids naturally occurring in algae and crustaceans with powerful free radical-scavenging and antioxidant effects. CX is a ketocarotenoid, which widely exists in marine animals, algae and a few terrestrial plants. CX has nine conjugated “C=C” in the center of the molecule [18], and it has been used in chickens to improve the laying rate or quality with some similar effects to other carotenoids, such as astaxanthin, lycopene and lutein [19,20,21,22]. As a lipophilic antioxidant, CX can scavenge reactive oxygen species and quench singlet oxygen [23]. So, it was found that CX could effectively help reduce oxidative reactions in tissues and the chick embryonic tissues [24] and significantly improve the egg production of laying hens [25]. Studies have shown that adding CX to the diet can improve the egg-production rate of poultry; as a feed additive, CX can significantly improve the egg-production rate and egg weight of poultry [26,27]. Adding CX to the diet could significantly increase the egg weight of ISA brown laying hens, and the egg weight increases with the increase of the content of cantharidin in the diet [27], and there is a significant increase in the egg production and average egg weight of 18-week-old ISA laying hens [25]. Moreover, under normal temperature conditions, studies have shown that adding cantharidin to breeder diets can significantly improve the antioxidant properties of egg yolks and chicks. Zhang et al. found that adding 6 mg/kg of cantharidin to the diet could significantly improve the antioxidant capacity of the serum and embryo of 23-week-old Sanhuang chickens [28]. Surai et al. found that adding 6 mg/kg or 12 mg/kg of cantharidin to the diet could reduce the liver malondialdehyde (MDA) level of broiler breeders [29]. These all suggested that Canthaxanthin could be used as an excellent feed additive to maintain the physiological state and laying performance of laying hens.

Huaixiang chicken is a famous Chinese indigenous, yellow-feathered, slow-growing broiler breed that has become gradually favored by consumers because of the excellent meat quality [30,31,32]. However, its low egg-production rate has affected the breeding and breed-promotion of Huaixiang chickens. It is necessary to develop new techniques and methods to improve the fertility index of Huaixiang hens, such as follicle development and egg-production rate. Therefore, this study took Huaixiang breeding chickens as test subjects to explore the mechanism of cantharidin on antioxidant performance, ovarian tissue structure, the expression of reproductive hormones and related receptors, follicle development, ovulation, and laying rate of Huaixiang breeding chickens under a normal-temperature environment, so as to provide a theoretical basis and data support for the application of cantharidin in the production of Huaixiang breeding chickens and subsequent studies.

## 2. Materials and Methods

### 2.1. Experimental Design, Animals and Diet

The Canthaxanthin(CX) was obtained from a Chinese company (batch no. UE01611012; DSM Ltd., Shanghai, P. R. China). All hens were 26 weeks old and living in an average environmental temperature of 25 ± 2 °C with a relative humidity of 65–75%. A total of 180 hens were randomly divided into 5 groups with six replicates of six hens each. Dietary treatment groups were as follows: (i) BD, basal diet, (ii) BD + 4 mg/kg CX(NT1), (iii) BD + 6 mg/kg CX(NT2), (iv) BD + 8 mg/kg CX(NT3), and (v) BD + 10 mg/kg CX(NT4). The schematic diagram of the experimental design is shown in Figure 1.

The experimental period lasted for 9 weeks. The nutrient and ingredients levels of basic diet are presented in Table 1, according to Dietary nutrient requirements of breeding hens in NY/T 3645-2020, the Agricultural industry standard of the People’s Republic of China. Incandescent light was used to illuminate chicken house for the whole period. A photo-period of 16 h/day from 7:00 am to 11:00 pm was controlled with light intensity of 10–15 Lx. Hens were free to access water and feed. The ingredients and nutrient levels of basic diet are presented in Table 2.

### 2.2. Sampling Collection

Blood samples were collected from wing vein in clean centrifuge tubes from 6 hens in each group at the 3rd, 6th and 9th weeks of experiments, respectively. Each sample was kept in two different tubes (with EDTA as an anticoagulant). The blood samples were centrifuged at 3000 rpm for 10 min at 4 °C for serum collection. Serum was stored in 2 mL plastic tubes at −80 °C for reproductive hormones and antioxidant index analysis. Ovary samples were collected for analysis of histomorphology, antioxidants, and reproductive hormones (*n* = 6 per treatment).

### 2.3. Analysis of Laying Performance

During the 9 weeks of experimental period, laying performance of the chickens were recorded including egg weight, egg rate and cracked egg rate per replicate. In addition, feed intake, the total number of egg numbers and average egg mass per bird was calculated. The average daily feed intake (g/d.), egg-production rate (%) and average egg weight (g) were calculated by repeating each week.

Average daily feed intake(ADFI) (g/d. birds) = total daily feed intake/number of chickens;

Laying rate(LR) (%) = total eggs per day/number of chickens;

Average egg weight(AEW) (g) = total egg weight/number of eggs laid;

Cracked egg rate(CER) (%) = broken eggs/total eggs.

### 2.4. Follicular Development and Ovarian Histomorphology

After 3, 6 or 9 weeks of supplement CX in the diet, the 6 ovaries collected from each group were weighed, then the follicles with a diameter of more than 1 mm on the ovary were removed with eye forceps. Subsequently, the diameter of the removed follicles was measured with an electronic vernier caliper. The grade follicles (>12 mm), large yellow follicles (LYF, 8–12 mm), small yellow follicles (SYF, 6–8 mm), large white follicles (LWF, 2–6 mm) and small white follicles (SWF, <2 mm) at various levels were counted.

### 2.5. Reproductive Hormones Analysis in Serum and Ovary

The concentration of reproductive hormones, including FSH, LH, E_2_ and Prog in serum and ovary, were determined using commercially available Enzyme Linked Immunosorbent Assay (ELISA) kits (MM-051802, MM-162302, MM-212302 and MM-114302) (Meimian industrial Co., Ltd., Nanjing, China). The sample pretreatment was as follows: Serum was defrosted at −80 °C and supernatant was centrifuged at 10,000 rpm at 4 °C for detection; ovarian tissue samples were removed from −80 °C and thawed at 4 °C, then broken by ultrasound at low temperature, and then centrifuged at 4 °C and 10,000 rpm to take supernatant for detection.

### 2.6. Antioxidant Index Analysis in Serum and Ovary

The concentration of antioxidant index including superoxide dismutase (SOD), glutathione peroxidase (GSH-Px), total antioxidant capacity (T-AOC), and malondialdehyde (MDA) were determined using commercially available Enzyme Linked Immunosorbent Assay (ELISA) kits(A001-3-2, A005-1-2, A015-2-1 and A003-1-2) (Jiancheng Bioengineering Research Institute Co., Ltd., Nanjing, China). The sample pretreatment was as described in Section 2.5.

### 2.7. Statistical Analysis

Data were expressed as means. Variability in data was expressed as standard error of means (SEM). Differences among means were tested by using Duncan’s test. Orthogonal polynomial contrasts were used to test the linear and quadratic effects. All data were analyzed with SAS 9.4 software (SAS Institute Inc., Cary, NC, USA). *p* < 0.05 was considered to be statistically significant.

## 3. Results

### 3.1. Laying Performance

Dietary CX supplementation significantly increased the ADFI and LR of the chickens after 3, 6 and 9 weeks of the experiment (*p* < 0.05, Table 3). There was a linear and quadratic improvement (*p* < 0.05) in ADFI at 3 weeks and AWE at 9 weeks, and a quadratic improvement (*p* < 0.01) in LR at 3, 6, 9 weeks associated with the increasing level of CX. CER was decreased with the concentration of CX ascending (*p* < 0.05), while the AEW was significantly increased at 7–9 weeks of the experiment (*p* < 0.05).

### 3.2. Reproductive Hormones in Serum

The serum hormone concentration of the chickens is shown in Table 4. There was a linear and quadratic decreasing trend (*p* < 0.05) in the FSH and LH of the chickens from 3 to 9 weeks, associated with the increasing level of CX, except for FSH at 9 weeks (*p* < 0.01). Meanwhile, Prog showed a quadratic downtrend at 9 weeks (*p* < 0.01). CX supplementation significantly improved FSH and LH concentrations in the 3rd, 6th and 9th weeks (*p* < 0.01), while Prog only increased after 9 weeks of experiments (*p* < 0.01). Dietary CX significantly improved E_2_ concentrations after 3 and 6 weeks (*p* < 0.01).

### 3.3. Reproductive Hormones of Ovary

As shown in Table 5, ovarian FSH at 3 weeks and LH at 6 weeks exhibited quadratic trends, while LH at 3 weeks showed linear and quadratic trends with the increasing of CX in the diet. CX significantly increased the FSH (*p* < 0.05) and LH (*p* < 0.01) of chicken ovaries after 3 weeks of supplementation. Additionally, dietary CX improved ovarian LH (*p* < 0.01) levels at the 6th week, whereas, the four reproductive hormones tended to increase (*p* > 0.05) at the 9th week.

### 3.4. Follicular Development

The effects of dietary CX supplementation on the follicular development of hens are presented in Table 6. There was a linear improvement (*p* < 0.01) in the LWF of chickens at 3 and 6 weeks, and a quadratic decrease (*p* < 0.01) in SYF associated with the increasing level of CX. The number of the LWF tends to increase as the CX concentration ascends at the three time points (*p* < 0.05), but the ovary weight and other types of follicle numbers were not affected (*p* > 0.05).

### 3.5. Serum Anti-Oxidant Indexes Analysis

There were linear and quadratic changes (*p* < 0.01) in the antioxidant indexes with the CX concentration ascending (Table 7). Antioxidant metabolite activities were determined in the serum of the chickens. Compared with the control group, the addition of CX can effectively increase the concentration of SOD and GSH-Px (*p* < 0.01) in the serum of chickens at 3, 6 and 9 weeks, but reduce the content of MDA (*p* < 0.01). There was a tendency to increase serum T-AOC (*p* > 0.05).

### 3.6. Ovary Antioxidant Indexes Analysis

There were linear and quadratic changes in SOD, GSH-Px and MDA (*p* < 0.01, Table 8). The addition of CX effectively increased the activities of ovarian SOD and GSH-Px (*p* < 0.01), but it decreased their content of MDA (*p* < 0.01). There was a tendency to increase ovarian T-AOC (*p* > 0.05).

## 4. Discussion

In this study, we focused on the effects of cantharidin on the laying rate and follicle development of Huaixiang hens, and detected the secretion and expression levels of key hormones and their receptors that affect follicle development and egg production in serum and ovarian tissue. The results of this study confirmed for the first time that dietary canthaxanthin could promote the laying rate and follicle development of Huaixiang laying hens. This study has laid a foundation for the application of cantharidin in breeding (in particular, the breeding of Huaixiang chickens) and provided reliable reference data.

### 4.1. Laying Performance

Varieties of research have shown that carotenoids functioned on poultry’s laying performance. In this study, we found that dietary CX supplementation significantly improved the laying rate of Huaixiang hens. Both the laying rate and average egg weight increased with the ascent of the CX concentration. A minimum CX supplementation of 6 mg/kg was enough to improve the laying performance. When the supplementation reached 10 mg/kg, the average daily feed intake of Huaixiang hens increased too. Previous studies also indicated that CX in the diet leads to better egg production than in non-supplemented groups [33]. A dietary supplementation of 6 mg/kg CX increased egg production in hens [34], which was consistent with the results of this study. As we know, carotenoids, such as β-carotene, can promote the synthesis of egg yolk precursors. CX and β-carotene have similar functions, both of which can increase egg yolk weight [27]. β-carotene stimulates the expression of estrogen enzymes in vitro, and in eggs, estrogen further regulates the synthesis of yolk lipids and proteins by stimulating gene expression in the liver to synthesize very low-density lipoprotein and vitellogenin [35,36]. Therefore, it is speculated that CX increased the yolk weight of poultry eggs possibly through a similar pathway.

### 4.2. Serum and Ovarian Reproductive Hormones

The hypothalamic–pituitary–gonad (HPG) axis regulates animal reproductive activities. Animal LH and FSH are involved in the follicle’s development and steroid hormones synthesis. Insufficient E_2_ secretion in the body will adversely affect follicular development and maturation, increase follicular atresia, and affect the laying performance of poultry [37]. Studies in mice had found that astaxanthin can act on the hypothalamus and ovary of mice, alleviate the adverse effects of natural aging on the hypothalamus and ovary, and improve fertility [38]. Both CX and astaxanthin are carotenoids. In this study, supplementation with CX improved the LH, FSH, E_2_ and Prog levels of Huaixiang broiler breeders, suggesting that CX may act as a regulator in the secretion of reproductive hormones, hence promoting ovarian maturation and improving the laying performance of breeders.

### 4.3. Follicular Development

Follicle development is important in determining the laying rate of poultry. Generally, follicle development in poultry consists of five steps: follicle recruitment, selection, development, maturation, and ovulation [39]. Ovulation occurs in sequence from large to small, and once the ovulation process occurs, a new pre-grade follicle will develop into a grade follicle [40]. It has been reported that cis-CX was isolated from the ovaries of female Artemia, suggesting that CX may play a role in female reproductive activity and embryonic development [41]. It is worth noting that the addition of 6 mg/kg CX increased the number of SYF, LYF and grade follicles in the ovaries of Huaixiang hens, indicating that CX can benefit follicle development. However, the mechanism needs to be further clarified.

### 4.4. Serum and Ovary Antioxidant Indexes

It has been documented that CX has free radical-scavenging properties [42]. CX can remove free radicals and absorb excess energy from highly reactive oxygen reactive species by altering the prooxidative/antioxidative balance [43,44].

In this experiment, we found that adding CX increased the activities of the antioxidant enzymes SOD and GSH-Px in the serum and ovary of Huaixiang chickens and reduced the content of malondialdehyde (MDA). In this experiment, the hens were at their peak of egg production and were exposed to oxidative stress likely from ovulation. It was speculated that CX could alleviate the adverse effects of oxidative stress. Adding CX together with “25-OH-D3” to the diet increased the T-AOC of the chicken liver, reduced the MDA content, and improved the antioxidant capacity of the breeding chickens [45]. In addition, the serum antioxidant capacity of hens was higher than that of the non-supplemented hens when 6 mg CX/kg was added in the feed [28]. As an effective fat-soluble antioxidant, CX can improve the body’s antioxidant enzyme activity, which has also been confirmed in mice [46]. It has been reported that ROS is produced during the ovulation process of laying hens; continuous large-scale ovulation in high-yielding laying hens leads to the accumulation of oxidative damage in the ovary and liver, and at the same time inhibits the expression of gonadotropin receptors in granulose cells [47,48]. This may be the reason for the rapid decline of ovarian function in hens.

## 5. Conclusions

This study indicates that the supplementation of CX in their diet improved the laying rate, promoted the ovulation process and maintained the reproductive hormones of hens by improving the antioxidant levels in their serum and ovaries. Being fed a basal diet supplemented with 6 mg/kg of CX brings out the best in the Huaixiang chickens; the laying rate and large white follicles (LWF) number of the hens, the concentration of reproductive hormones(LH, FSH, E_2_ and Prog), and the activities of anti oxidant enzymes (SOD and GSH-Px) presented the best data and states, while the content of lipid peroxidation product MDA presented its lowest value.

## Figures and Tables

**Figure 1 biology-12-01375-f001:**
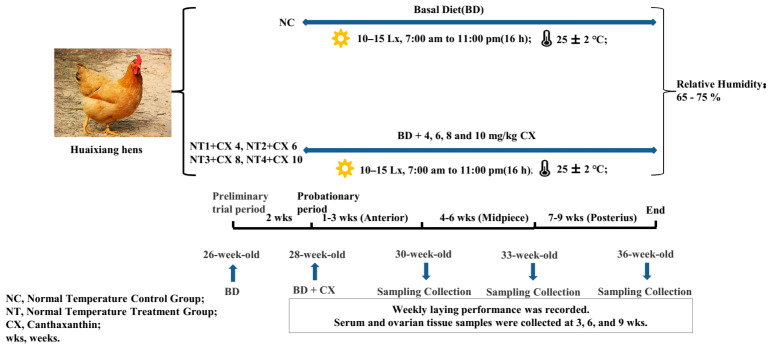
The schematic diagram of the experimental design.

**Table 1 biology-12-01375-t001:** The control and treatment groups in this study.

group	Diet
NC	basal diet(BD)
NT1	BD + 4 mg/kg CX
NT2	BD + 6 mg/kg CX
NT3	BD + 8 mg/kg CX
NT4	BD + 10 mg/kg CX

**Table 2 biology-12-01375-t002:** Basal diet composition of this experiment.

Item	Contents (%)
Corn	55
Soybean meal	20
Wheat bran	9.5
Fish meal	5.0
Limestone	7.5
CaHPO_4_	2.5
NaCl	0.1
Premix ^1^	0.4
Nutrient levels ^2^	
ME/(MJ/kg) ^2^	11.6
CP	15.5
Ca	2.0
TP	0.63
Met	0.40
Cys	0.30
Lys	0.80

^1^ Premix provided per kg of diet: VA 9000 IU, VD 2500 IU, VE 20 IU, VB 1212 μg, VK 2.4 mg; Mn100 mg, Zn 60 mg, Fe 25 mg, Cu 5 mg, Co 0.1 mg(Mn, Zn, Fe, Cu, Co were provided in the form of sulPhate), Se (N_2_SeO_3_·5H_2_O) 0.2 mg, I(KI) 0.5 mg; ^2^ Nutrient levels on DM basis; except for metabolic energy (ME), others are measured values; ME calculated according to Chinese feed ingredient database.

**Table 3 biology-12-01375-t003:** Effects of CX on the laying performance of Huaixiang hens.

							*p*-Value	
Items	CON	NT1	NT2	NT3	NT4	SEM	Linear	Quadratic
ADFI(g/d)								
3W	109.18 ^b^	108.62 ^bc^	113.55 ^a^	108.64 ^bc^	106.24 ^c^	0.75	0.033	0.001
6W	101.41 ^b^	103.97 ^b^	103.91 ^b^	105.35 ^b^	112.77 ^a^	2.23	0.007	0.209
9W	104.62 ^c^	106.55 ^bc^	109.90 ^bc^	114.74 ^b^	125.12 ^a^	2.72	<0.001	0.101
LR(%)								
3W	65.00 ^b^	70.50 ^a^	71.76 ^a^	66.17 ^b^	66.67 ^b^	1.16	0.788	<0.001
6W	53.11 ^b^	57.22 ^b^	61.11 ^a^	56.67 ^b^	55.74 ^b^	1.93	0.447	<0.001
9W	53.05 ^c^	56.52 ^b^	60.00 ^a^	54.44 ^bc^	54.92 ^bc^	1.02	0.611	<0.001
AEW(g)								
3W	41.57 ^ab^	41.23 ^ab^	42.03 ^a^	41.08 ^b^	40.90 ^b^	0.28	0.102	0.186
6W	44.75 ^ab^	44.17 ^b^	45.51 ^a^	45.43 ^a^	44.13 ^b^	0.40	0.986	0.066
9W	45.70 ^c^	44.13 ^bc^	46.15 ^a^	47.03 ^abc^	46.42 ^ab^	0.25	0.007	0.043
CER(%)								
3W	0.43	0.42	0.34	0.43	0.42	0.04	0.987	0.314
6W	0.45 ^a^	0.36 ^bc^	0.35 ^c^	0.44 ^a^	0.42 ^a^	0.02	0.733	0.011
9W	0.54	0.52	0.44	0.49	0.49	0.06	0.547	0.501

NC; Negative control; NT1 (4 mg/kg), canthaxanthin supplemented diet 4, NT2 (6 mg/kg), canthaxanthin supplemented diet 6, NT3 (8 mg/kg), canthaxanthin supplemented diet 8, NT4 (10 mg/kg), canthaxanthin supplemented diet 10; ADFI, Average Daily Feed Intake; LR, Laying Rate; AEW, Average Egg Weight; CER, Cracked egg rate. ^abc^ Different superscript letters in the same line indicate significant differences (*p* < 0.05).

**Table 4 biology-12-01375-t004:** Effects of CX on serum reproductive hormone levels of Huaixiang hens.

							*p*-Value	
Items	CON	NT1	NT2	NT3	NT4	SEM	Linear	Quadratic
FSH/(mIU/mL)								
3W	28.43 ^b^	33.18 ^a^	33.74 ^a^	30.15 ^b^	28.33 ^b^	0.57	0.101	<0.001
6W	22.85 ^c^	32.91 ^a^	33.43 ^a^	31.07 ^ab^	29.85 ^b^	0.89	0.002	<0.001
9W	20.35 ^c^	25.48 ^b^	26.28 ^b^	28.78 ^a^	29.04 ^a^	0.49	<0.001	0.001
LH/(ng/mL)								
3W	176.17 ^c^	219.5 ^b^	234.45 ^a^	221.04 ^b^	217.89 ^b^	2.96	<0.001	<0.001
6W	177.07 ^c^	221.41 ^ab^	228.05 ^a^	207.26 ^b^	204.96 ^b^	5.23	0.031	0.001
9W	178.60 ^d^	223.16 ^ab^	226.48 ^a^	212.64 ^bc^	208.11 ^c^	3.60	0.002	<0.001
E_2_/(pg/mL)								
3W	410.18 ^b^	420.36 ^b^	512.59 ^a^	423.61 ^b^	411.93 ^b^	9.81	0.833	0.001
6W	402.60 ^b^	458.44 ^a^	463.61 ^a^	442.97 ^ab^	436.36 ^ab^	15.46	0.312	0.026
9W	361.48	373.27	367.94	363.43	362.34	15.45	0.872	0.676
Prog/(ng/mL)								
3W	28.90	28.43	28.17	27.03	26.48	1.78	0.293	0.880
6W	30.28	32.6	31.49	30.86	30.93	1.26	0.917	0.414
9W	28.18 ^b^	32.63 ^a^	33.12 ^a^	29.67 ^b^	27.69 ^b^	0.79	0.143	<0.001

NC; Negative control; NT1 (4 mg/kg), canthaxanthin supplemented diet 4, NT2 (6 mg/kg), canthaxanthin supplemented diet 6, NT3 (8 mg/kg), canthaxanthin supplemented diet 8, NT4 (10 mg/kg), canthaxanthin supplemented diet 10. FSH, follicle-stimulating hormone; LH, Luteinizing Hormone; E_2_, estrogen; Prog, progestogen. ^a,b,c,d^ Different superscript letters in the same line indicate highly significant differences (*p* < 0.01).

**Table 5 biology-12-01375-t005:** Role of CX on ovarian reproductive hormone of Huaixiang hens.

							*p*-Value	
Items	CON	NT1	NT2	NT3	NT4	SEM	Linear	Quadratic
FSH/(mIU/mL)								
3W	21.62 ^b^	22.97 ^ab^	25.52 ^a^	21.98 ^b^	21.92 ^b^	1.04	0.912	0.046
6W	19.44	22.58	23.08	19.56	19.24	2.01	0.603	0.177
9W	18.62	21.83	23.76	19.56	18.65	1.74	0.698	0.052
LH/(ng/mL)								
3W	174.27 ^b^	209.05 ^a^	222.28 ^a^	211.82 ^a^	211.33 ^a^	4.58	<0.001	<0.001
6W	163.12 ^b^	186.21 ^ab^	210.22 ^a^	185.58 ^ab^	184.70 ^ab^	10.28	0.220	0.031
9W	161.59	200.52	189.68	179.30	166.30	11.60	0.918	0.117
E_2_/(pg/mL)								
3W	566.13	576.27	577.55	571.79	570.27	18.23	0.949	0.666
6W	625.89	640.48	652.15	628.96	627.50	17.97	0.887	0.343
9W	546.95	556.39	554.47	552.36	548.71	14.58	0.992	0.639
Prog/(ng/mL)								
3W	32.05	33.03	32.00	30.15	29.14	1.41	0.079	0.383
6W	27.76	27.59	31.20	28.77	27.70	1.27	0.796	0.130
9W	27.98	30.62	30.75	27.73	28.65	1.16	0.681	0.161

NC; Negative control; NT1 (4 mg/kg), canthaxanthin supplemented diet 4, NT2 (6 mg/kg), canthaxanthin supplemented diet 6, NT3 (8 mg/kg), canthaxanthin supplemented diet 8, NT4 (10 mg/kg), canthaxanthin supplemented diet 10. FSH, follicle-stimulating hormone; LH, Luteinizing Hormone; E_2_, estrogen; Prog, progestogen. ^a,b^ Different superscript letters in the same line indicate highly significant differences (*p* < 0.01).

**Table 6 biology-12-01375-t006:** Effects of CX on follicular development of Huaixiang hens.

							*p*-Value	
Items	CON	NT1	NT2	NT3	NT4	SEM	Linear	Quadratic
Ovary weight/g								
3W	3.25	3.26	3.36	3.29	3.28	0.23	0.898	0.809
6W	3.37	3.55	3.4	3.39	3.38	0.42	0.913	0.875
9W	3.21	3.41	3.52	3.45	3.31	0.20	0.714	0.291
GF/fcs								
3W	6.33	5.83	6.17	5.83	6.17	0.39	0.789	0.498
6W	5.67	6.33	6.50	6.17	6.00	0.44	0.722	0.199
9W	5.00	5.67	6.17	5.50	5.50	0.46	0.568	0.155
SYF/fcs								
3W	12.83	13.67	13.83	13.50	13.33	1.27	0.838	0.605
6W	14.83 ^ab^	17.17 ^a^	17.33 ^a^	16.33 ^ab^	13.83 ^a^	1.03	0.392	0.010
9W	12.33	13.67	15.00	13.50	13.33	0.87	0.511	0.085
LYF/fcs								
3W	2.33	2.50	2.83	2.50	3.33	0.36	0.195	0.537
6W	2.17	2.67	2.50	2.17	1.83	0.32	0.259	0.138
9W	2.00	1.50	2.00	1.83	1.33	0.30	0.305	0.560
SWF/fcs								
3W	39.67	40.33	41.00	40.00	39.33	1.32	0.813	0.389
6W	30.33	33.00	33.33	32.67	32.33	1.57	0.468	0.246
9W	26.00	27.33	27.67	27.33	26.67	1.47	0.776	0.404
LWF/fcs								
3W	33.67 ^b^	38.33 ^b^	37.33 ^b^	54.00 ^a^	47.67 ^a^	3.08	<0.001	0.711
6W	30.33 ^b^	35.67 ^ab^	35.50 ^ab^	34.67 ^ab^	40.00 ^a^	1.94	0.006	0.928
9W	41.67	44.00	43.67	54.00	42.67	2.4	0.136	0.083

NC; Negative control; NT1 (4 mg/kg), canthaxanthin supplemented diet 4, NT2 (6 mg/kg), canthaxanthin supplemented diet 6, NT3 (8 mg/kg), canthaxanthin supplemented diet 8, NT4 (10 mg/kg), canthaxanthin supplemented diet 10. GF, Grade Follicles; SYF, Small Yellow Follicles; LYF, Large Yellow Follicles; SWF, Small White Follicles; LWF, Large White Follicles; fcs, Follicles Per section. ^a,b.^ Different superscript letters in the same line indicate significant differences (*p* < 0.05).

**Table 7 biology-12-01375-t007:** Effects of CX on serum antioxidant indexes of Huaixiang hens.

							*p*-Value	
Items	CON	NT1	NT2	NT3	NT4	SEM	Linear	Quadratic
SOD/(U/mL)								
3W	431.55 ^c^	454.72 ^c^	556.45 ^a^	507.28 ^b^	498.92 ^b^	11.54	<0.001	<0.001
6W	331.05 ^c^	416.86 ^b^	461.33 ^a^	474.73 ^a^	447.57 ^ab^	12.64	<0.001	<0.001
9W	310.95 ^b^	444.97 ^a^	460.81 ^a^	458.55 ^a^	442.83 ^a^	14.56	<0.001	<0.001
T-AOC/(U/mL)								
3W	15.03 ^b^	16.45 ^ab^	17.08 ^a^	16.09 ^ab^	15.27 ^ab^	0.59	0.949	0.019
6W	16.67	17.52	17.59	17.16	16.90	0.76	0.972	0.362
9W	14.69 ^a^	16.33 ^b^	15.99 ^b^	17.33 ^c^	15.37 ^a^	0.99	<0.001	<0.001
GSH-Px/(U/mL)								
3W	67.91 ^d^	90.69 ^c^	111.61 ^a^	103.39 ^ab^	100.45 ^b^	2.89	<0.001	<0.001
6W	69.34 ^c^	98.14 ^b^	111.32 ^a^	107.84 ^ab^	107.43 ^ab^	3.60	<0.001	<0.001
9W	65.33 ^d^	84.24 ^c^	109.46 ^a^	93.50 ^b^	95.72 ^b^	2.73	0.469	0.164
MDA/(nmol/L)								
3W	8.09 ^a^	5.87 ^b^	4.51 ^c^	4.66 ^c^	4.65 ^c^	0.22	<0.001	<0.001
6W	7.65 ^a^	5.14 ^b^	4.81 ^b^	4.96 ^b^	5.37 ^b^	0.20	<0.001	<0.001
9W	8.89 ^a^	6.45 ^bc^	5.87 ^c^	6.92 ^b^	7.31 ^b^	0.26	0.009	<0.001

NC; Negative control; NT1 (4 mg/kg), canthaxanthin supplemented diet 4, NT2 (6 mg/kg), canthaxanthin supplemented diet 6, NT3 (8 mg/kg), canthaxanthin supplemented diet 8, NT4 (10 mg/kg), canthaxanthin supplemented diet 10. SOD, superoxide dismutase; GSH-Px, glutathione peroxidase; T-AOC, total antioxidation capacity; MDA, malondialdehyde. ^a,b,c,d^ Different superscript letters in the same line indicate highly significant differences (*p* < 0.01).

**Table 8 biology-12-01375-t008:** Effects of CX on ovary antioxidant indexes of Huaixiang hens.

							*p*-Value	
Items	NC	NT1	NT2	NT3	NT4	SEM	Linear	Quadratic
SOD/(U/mL))								
3W	684.84 ^b^	773.51 ^a^	794.02 ^a^	787.99 ^a^	787.96 ^a^	22.64	0.012	0.037
6W	635.86 ^b^	723.94 ^a^	763.15 ^a^	757.02 ^a^	759.81 ^a^	20.29	0.001	0.017
9W	608.23 ^b^	724.37 ^a^	746.69 ^a^	738.25 ^a^	761.17 ^a^	15.99	<0.001	0.005
T-AOC/(U/mL)								
3W	23.71	24.52	25.04	23.76	24.06	1.17	0.985	0.533
6W	29.87	29.11	31.21	30.14	30.85	0.73	0.227	0.931
9W	29.09	28.96	30.41	30.24	27.79	0.96	0.674	0.113
GSH-Px/(U/mL)								
3W	133.63 ^b^	153.91 ^a^	171.74 ^a^	169.15 ^a^	168.07 ^a^	5.60	<0.001	0.013
6W	119.22 ^c^	156.46 ^b^	160.63 ^ab^	165.52 ^a^	159.94 ^ab^	2.23	<0.001	<0.001
9W	117.14 ^b^	154.39 ^a^	165.89 ^a^	157.26 ^a^	157.04 ^a^	4.83	<0.001	<0.001
MDA/(nmol/L)								
3W	6.49 ^a^	4.18 ^b^	3.97 ^b^	4.03 ^b^	4.01 ^b^	0.30	<0.001	0.001
6W	6.35 ^a^	3.62 ^b^	3.37 ^b^	3.42 ^b^	3.50 ^b^	0.25	<0.001	<0.001
9W	5.80 ^a^	3.38 ^b^	3.37 ^b^	3.39 ^b^	3.51 ^b^	0.26	<0.001	<0.001

NC; Negative control; NT1 (4 mg/kg), canthaxanthin supplemented diet 4, NT2 (6 mg/kg), canthaxanthin supplemented diet 6, NT3 (8 mg/kg), canthaxanthin supplemented diet 8, NT4 (10 mg/kg), canthaxanthin supplemented diet 10. SOD, superoxide dismutase; GSH-Px, glutathione peroxidase; T-AOC, total antioxidation capacity; MDA, malondialdehyde. ^a,b,c^ Different superscript letters in the same line indicate highly significant differences (*p* < 0.01).

## Data Availability

Data is contained within the article.

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
