# Peer review of "Dietary Canthaxanthin Supplementation Promotes the Laying Rate and Follicular Development of Huaixiang Hens"

_biology, 2023, doi:10.3390/biology12111375_

Round 1

Reviewer 1 Report

The study titled "Dietary Canthaxanthin Supplementation Promotes the Laying Rate and Follicular Development of Huaixiang Hens" explores the effects of adding canthaxanthin to the diet of Huaixiang hens. The researchers aimed to investigate whether this dietary supplementation could enhance egg-laying productivity and improve the development of ovarian follicles in these hens. The results of the study indicate that the inclusion of canthaxanthin in the hens' diet led to positive outcomes. The laying rate of the hens increased, suggesting improved egg production. Additionally, the researchers observed enhanced development of ovarian follicles, which are essential structures for egg maturation. However, the paper need extensive revision before publication. The following changes must be considered. 

In the abstract the abbreviations must be defined first. and which group produced the best results must be identified. The authors probably do not know how to abbreviate a word where it should be presented for the first time. The introduction part is very small and does not define well the background of the study. age of the birds must be define. the starting date of the experiment must be defined. abbreviations in the tables must be defined. the conclusion of the study is general and must be clarified. 

Author Response

Dear Reviewer,

On behalf of my co-authors, we thank you very much for giving us the opportunity to revise our manuscript. We appreciate Editors and Reviewers very much for the positive and constructive comments and suggestions on our manuscript entitled “Dietary Canthaxanthin Supplementation Promotes the Laying Rate and Follicular Development of Huaixiang Hens (Manuscript ID: Biology-2555353). We have studied comments carefully and revised portions needed as the changes are tracked in the current manuscript. We sincerely appreciate your consideration of our manuscript, and we look forward to receiving comments after you done assessing the response given. Please, if you have any queries or any other thing for me to do to improve on this manuscript, do well to contact me.

The main corrections in our paper and response to your comments, Please see the attachment.

Reviewer 2 Report

Simple summary

hat mainly exist in the chloroplasts. What does mean "hat"?

Abstract

Write this "CX" abbreviation beside the full name first.

Please, replace chickens with hens

Delete of hens from this sentence "white follicles (LWF) number of hens"

"Dietary CX increased the activities of anti-oxidant enzymes SOD and GSH-Px" rephrase this part of this sentence.

Where is the recommended level of CX ?

Introduction

do not use full name " Canthaxanthin" after using the abbreviation "These all suggested that Canthaxanthin could be used as an excellent"

Please, replace chickens with hens in all manuscript.

as authors mentioned that " Huaixiang chicken is a famous Chinese indigenous yellow-feathered slow-growing broiler" therefore, did authors used in this experiment yellow-feathered slow-growing broiler breeders or hens?

Why authors wrote their findings in the end of the introduction section?

Materials and methods

"A total of 180 hens" where is the age of hens?

Authors formulated the basal diet based on what? Did the basal diet cover these breed requirements?

Based on what authors chose CX levels?

How authors raised these hens in floor or cages?

Did authors calculate feed conversion ratio? if the answer yes, add them, but if the answer no why?

How authors calculate average daily feed intake?

Did authors evaluate egg quality?

Why authors collected the samples at different time after 3, 6, and 9 wk of the experimental period? Did authors consider the time effect in their statistical analysis?

Write how authors determine these hormones "FSH, LH, E2 and P4" in ovary? How authors extracted these hormones "FSH, LH, E2 and P4" from ovary?

How authors extracted antioxidant enzymes from ovary?

Write their full name first "SOD, GSH-Px, T-AOC and MDA"

Results

Write the full name first "LR AWE "

Discussion

Did authors determine β- carotene in the diet, serum, or egg?

Did authors determine very low-density lipoprotein and vitellogenin?

Did authors determine CX in the diet, serum, liver, or egg?

conclusion

Where is the recommended level in Huaixiang Hens?

Table 6

P value was 0.083, therefore the findings not significant, so delete significant letters.

Table 7

Add significant letters to T-AOC in 9w

Author Response

(The authors gave the same response as above.)

Reviewer 3 Report

Dear authors, 

The manuscript was well-written and the content was informative and well-presented. I commend the authors for the comprehensive and systematic review of the topic. The manuscript will be a valuable contribution to this journal.

However, I’ve mentioned a few minor corrections that need to be corrected in the comment section of the main manuscript file. Some of these are the following:

Please add one line at the end of the abstract, which basically explains the basic output of this study and the future recommendations related to this study work as well.

In line 21: Please mention the hen's age here as well, whether they are day-old chicks or some weeks-old hens. 

In line 29: Please mention the inclusion level of CX more specifically in your conclusion statement. 

Please rewrite the conclusion part of this manuscript: To highlight the basic research gap that authors actually try to cover in this study along with their future recommendations, on the basis of their conclusion.

Please review the manuscript to ensure that there are no typographical errors or inconsistencies in formatting.

Please set the entire list of references according to "Biology" MDPI journal instructions.  

Best wishes 

Author Response

(The authors gave the same response as above.)

Round 2

Reviewer 1 Report

no comments

Author Response

Dear reviewer, thank you again for your sincere and constructive comments.

Reviewer 2 Report

Authors did not calculate feed conversion ratio, egg quality, and indices related to lipid metabolism. Also, authors failed to explain their results, therefore, they are finding are just observations.

Author Response

(The authors gave the same response as above.)
